# Arylamines QSAR-Based Design and Molecular Dynamics of New Phenylthiophene and Benzimidazole Derivatives with Affinity for the C111, Y268, and H73 Sites of SARS-CoV-2 PLpro Enzyme

**DOI:** 10.3390/ph17050606

**Published:** 2024-05-09

**Authors:** Gianfranco Sabadini, Marco Mellado, César Morales, Jaime Mella

**Affiliations:** 1Instituto de Química y Bioquímica, Facultad de Ciencias, Universidad de Valparaíso, Av. Gran Bretaña 1111, Valparaíso 2360102, Chile; gianfranco.sabadini@postgrado.uv.cl; 2Instituto de Investigación y Postgrado, Facultad de Ciencias de la Salud, Universidad Central de Chile, Santiago 8330507, Chile; 3Laboratorio de Materiales Funcionales, Centro Integrativo de Biología y Química Aplicada (CIBQA), Facultad de Ciencias de la Salud, Universidad Bernardo OHiggins, General Gana 1702, Santiago 8320000, Chile; cesar.morales@ubo.cl; 4Centro de Investigación, Desarrollo e Innovación de Productos Bioactivos (CInBIO), Universidad de Valparaíso, Av. Gran Bretaña 1111, Valparaíso 2360102, Chile

**Keywords:** SARS-CoV-2, COVID-19, coronavirus, PLpro, QSAR, Free-Wilson, molecular dynamics, antiviral

## Abstract

A non-structural SARS-CoV-2 protein, PLpro, is involved in post-translational modifications in cells, allowing the evasion of antiviral immune response mechanisms. In this study, potential PLpro inhibitory drugs were designed using QSAR, molecular docking, and molecular dynamics. A combined QSAR equation with physicochemical and Free-Wilson descriptors was formulated. The r^2^, q^2^, and r^2^_test_ values were 0.833, 0.770, and 0.721, respectively. From the equation, it was found that the presence of an aromatic ring and a basic nitrogen atom is crucial for obtaining good antiviral activity. Then, a series of structures for the binding sites of C111, Y268, and H73 of PLpro were created. The best compounds were found to exhibit pIC_50_ values of 9.124 and docking scoring values of −14 kcal/mol. The stability of the compounds in the cavities was confirmed by molecular dynamics studies. A high number of stable contacts and good interactions over time were exhibited by the aryl-thiophenes Pred14 and Pred15, making them potential antiviral candidates.

## 1. Introduction

At the beginning of 2020, the existence of a new beta-coronavirus in the city of Wuhan, China, nicknamed 2019-nCoV, was published. By March 2024, a total of 704 million cases and 7 million deaths attributed to this new coronavirus have been reported worldwide [1]. At the end of 2020, the first vaccine against SARS-CoV-2 was published (BNT162b2 mRNA COVID-19 Vaccine) [2,3]. The latest report from 2023 announced 183 vaccines in the clinical phase and 199 in the preclinical phase [4]. Vaccines have shown that they are capable of inducing the formation of antibodies against SARS-CoV-2 [5,6,7], which makes them a good strategy for the prevention of new infections. However, other short-term strategies are needed to treat patients with severe conditions [8,9]. The use of potent antiviral drugs is clearly important in these cases. To date, there are antivirals reported against SARS-CoV-2: Remdesivir, Molnupiravir [10], Paxlovid [11], and Pemgarda (a recombinant human monoclonal IgG1λ antibody) [12]. Remdesivir is effective in reducing recovery time, but in critically ill patients there is no effect in reducing the mortality rate [13]. On the other hand, Molnupiravir generates a risk of tumor formation. In addition, this compound can generate mutations in the cells that generate sperm, which could directly affect the embryo [14]. Therefore, the generation of new drugs against SARS-CoV-2, which have optimal characteristics in terms of cytotoxicity, reduction of recovery time and reduction of the mortality rate, is a priority worldwide.

Among the potential molecular targets [15] in which an anti-coronavirus drug could act are 3-chymotrypsin-like protease (3CLpro) [16,17,18], Spike [19,20,21], RNA-dependent RNA polymerase (RdRp) [22,23,24], and papain like protease (PLpro) [25,26,27,28], among others. The PLpro enzyme has drawn the attention of the scientific community because it is involved in post-translational modifications in host cells, allowing evasion of antiviral immune response mechanisms [29,30]. PLpro is one of the 16 non-structural proteins of the virus. This is part of the polyproteins called pp1a and pp1ab, which originate from reading viral RNA. PLpro is separated from these polyproteins autocatalytically. PLpro has proteolytic capacity, therefore it can recognize and cut the LXGG amino acid sequence of the C-terminal chains of ubiquitins and of the ISG15 protein. It also recognizes these sequences in the pp1a and pp1ab proteins, separating the nsp1/2, nsp2/3 and nsp3/4 sites [31]. The consequence of acting as a protease at the C-terminus of ubiquitins and ISG15 is to prevent the formation of covalent complexes that trigger the antiviral response [32,33]. There are also reports where PLpro together with the SUD protein have a role in the stabilization of the RCHY1 domain of the E3 Protein [34]. The E3 protein interacts directly with p53 by polyubiquinating it, causing its immediate destruction [35]. Another E3 domain is MDM2, which can monoubiquitinate p53, leading to its exportation to the cytosol. This process inhibits apoptosis [36] and theoretically could promote cancer development. The recently described mechanisms are important for virus replication, since p53 inhibits its replication. Therefore, the development of compounds that inhibit PLpro would be a good antiviral strategy.

The amino acids responsible for the proteolytic capacity of PLpro are the catalytic triad of residues C111, H272 and D286 [30,37]. These are located between the “thumb and palm” subdomains (P1 site), although there are also another three domains: P2, P3 and P4. These four sites as a whole constitute the interaction region of the LXGG amino acids (X = asparagine, arginine or lysine) with the substrates, where glycines are located in P1 and P2 [38,39]. The binding sites that do not correspond to the active site (P2, P3 and P4) are composed of the following amino acids: Y268, M208, P247, P248, T301, P248, Y264, N267, Q269, L162, C270, G271 and Y273 [26].

Current known inhibitors of PLpro can act at two allosteric sites [40,41,42]. One of them is centered on Y268 [43] and the other on H73 [44]. Most of the reported inhibitors bind to the Y268 site. These mostly interact with the Bl2 loop of PLpro [43,45,46].

Prominent allosteric inhibitors of the Y268 site include XR8-23 (IC_50_ = 0.39 μM), XR8-24 (IC_50_ = 0.56 μM) [46], and GRL-0617 (IC_50_ = 2.4 μM) [47] (Figure 1). On the other hand, the inhibitors reported for the allosteric site that is centered on amino acid H73 prevent PLpro from binding to ISG15. Specifically, the Ser22, Met23 and Glu27 amino acids of ISG15 lose the ability to interact with PLpro [44]. Amino acid mutations in this allosteric region, such as F69A and V66A, reduce the enzymatic activity of PLpro. In addition, the enzyme kinetics slow down compared to its wild-type state [30]. The inhibitors reported to date for the H73 site are YRL (IC_50_ = 6.68 μM), HBA (IC_50_ = 3.99 μM), and HE9 (IC_50_ = 3.76 μM) (Figure 1) [44].

Up to now, more than 100 molecules with potential inhibitory characteristics of PLpro activity have been reported [13,26,37,43,46,47,48,49,50,51,52,53,54,55]. This large number of molecules available allows the appropriate search for predictive QSAR equations, useful for the design of new promising compounds. SAR studies have been reported on PLpro, which describe the characteristics that the new molecule should have, but only in qualitative terms [49,51]. There are no QSAR studies on PLpro for SARS-CoV-2. There is only one QSAR study for PLpro on SARS-CoV-1 [56], which used a set of 40 active molecules against PLpro. The descriptors used in this study were based on the SMILES codes of each molecule and graphic descriptors such as GAO (graph of atomic orbital), HSG (hydrogen-suppressed graph), and HFG (hydrogen-filled graph). The limitations of this study are the difficulties in interpreting the descriptors, the lack of statistical validation and of molecular dynamics simulations, and that it was carried out using an out-of-date database for SARS-CoV-1.

In the present study, we selected a library of 113 molecules [37,43,46,48,49,50,51,52,53,54,55,57] to be analyzed in a 2D QSAR study with two purposes: (a) design new molecules with anti-SARS-CoV-2 activity; and (b) predict the anti-SARS-CoV-2 activity of commercial antivirals. To validate the equations, we use statistical parameters such as: q^2^_F1–3_, MAE, and Y-random test, among others. To test the proposed molecules, we carried out docking and molecular dynamics studies, finding a good correlation between the results.

## 2. Results and Discussion

### 2.1. QSAR

The statistical parameters for internal and external validation of the QSAR model are presented in Table 1. The final equation was obtained by cross-validation using the PLS method. The optimal value of q^2^ was obtained using 5 components (q^2^ = 0.770). Figure 2 shows the distribution of experimental versus predicted activities for the training (Figure 2A) and test set (Figure 2B). As can be seen, there is a good distribution of values around the line *y = x*. Compounds are distributed along 3 logs units of activity (compound 77, least active, pIC_50_ = 3.903; compound 43, most active, pIC_50_ = 6.947). Table 2 shows all the values of experimental, predictive activity, and residual value for each compound. All compounds have no more than one log unit of residual value (Figure 2C, residual = pIC_50_ exp − pIC_50_ pred).

For the external validation, we carried out the calculation of the descriptors r^2^_test_, $q_{F1}^{2}$, $q_{F2}^{2}$, $q_{F3}^{2}$, MAE and CCC. The values can be seen in Table 1, and they are all within the allowed limits, which validates the proposed equation. Additionally, in order to rule out the possibility that the results are the consequence of random correlation, we carry out the randomization of all the values of biological activities (Y-random test), and we recalculate the values of q^2^ and r^2^. If the model is not the result of chance, the values obtained for q^2^ and r^2^ should be as low as possible. In our case, average values of 0.0003 and 0.118 were obtained for q^2^ and r^2^ respectively. For all the above, we can conclude that the equation has adequate external predictability.

Below we present the equation obtained from the 5-component PLS analysis:pIC_50_ = 3.84264 − 0.23411nF − 0.79971nCl − 0.21961nO + 0.16803ALogP + 0.04589ALogP^2^ + 0.00796AMR + 0.00119DPSA-3 + 0.2659nAr + 0.17442nBasic + 0.22252nAcid + 0.07801nHBDon + 0.00225TopoPSA + 0.00145VABC.
where, ALogP (lipophilicity), ALogP^2^ (the square of ALogP), AMR (molar refractivity), DPSA-3 (Difference of PPSA-3 and PNSA-3), nAr (number of aromatic rings), nBasic (number of basic nitrogen atoms), nAcid (number of carboxylic acids), nF (number of fluorine atoms), nCl (number of chlorine atoms), nHBDon (number of hydrogen bond donor atoms), nO (number of oxygen atoms), TopoPSA (topological polar surface area) and VABC (van der Waals volume with Atomic and Bond Contributions).

The descriptors that negatively contribute to biological activity, expressed as pIC_50_, are the number of fluorine, chlorine and oxygen atoms present in the structures. Therefore, it seems that, in order to achieve a good activity on PLpro, the presence of electronegative atoms and hydrogen bond acceptors should be avoided. Anionic amino acids such as Glu161, Asp164, Glu167, Asp302, and electron-rich residues, such as Tyr171, Tyr264, Tyr 268, Tyr273, Cys270, Met206, Met208, are found in the active site C111 and in the adjacent site Y268 of PLpro. Therefore, the resulting equation is consistent with the amino acid environment of these cavities. To achieve a good antiviral activity, molecules of an electrophilic nature will be preferred. It should be noted that the equation, however, does not consider the halogen-bonding effect, so the insertion of bromine or iodine could generate outlier compounds that do have good affinity for the cavity. An example of this anomaly can be found in the recent work by Chia-Chuan Cho et al., 2022 [13], where they report the polychlorinated compound TCID. Despite the presence of chlorine, the affinity of this compound for PLpro could be due to the formation of halogen bonds with the electron-rich residues of the enzyme.

Positively contributing descriptors are ALogP, AMR, DPSA-3, nAr, nBasic, nAcid, nHBDon, TopoPSA, and VABC. Of these, that with the highest contribution coefficient is nAr (0.2659); this could be because a greater number of aromatic rings favors pi-stacking interactions with the abundant tyrosines present (Tyr171, 264, 268 and 273) in PLpro, which we confirmed in our docking and molecular dynamics studies. As already explained, the contribution of the number of oxygen atoms is negative therefore, the number of basic nitrogen atoms becomes more important for the activity than the number of carboxylic acid groups. To achieve good activity, the structures should ideally contain a protonatable nitrogen that can perform hydrogen-bond interactions with Glu161, Asp164, Glu167, and Asp302. On the other hand, protonable nitrogen atoms can carry out pi-cation interactions with the tyrosine residues. This conclusion is reinforced by the fact that the number of hydrogen bond donors is favorable for activity. Finally, as the DPSA-3, TopoPSA and VABC descriptors have positive coefficients, then we can generalize that if the compound is easy to polarize, the better its anchorage to cavities. This is evidenced in the most active compounds of the studied series 41, 43, and 44, which have a large surface area and sulfur atoms in their structure, which contribute to the polarizability of the compounds.

### 2.2. Evaluation of the Model in Antivirals from the Market

In order to evaluate in silico the potential inhibitory capacity of PLpro of molecules already reported, we carried out pIC_50_ prediction for more than 60 compounds of diverse structures and pharmacological families, including 42 antivirals present on the market. The best predictions by the QSAR equation were obtained for five antivirals, which reported the best scoring values in docking: indinavir, etravirine, palinavir, lopinavir, and atazanavir (Figure 3). Indinavir in its two possible protonation states had the highest affinity at all three possible binding sites (active site (C111), Y268 site, and H73 site). The best score for Indinavir was −10.329 at the active site. The main interactions for indinavir at the active site were piperazine NH+, ionic interaction with Asp164, and pi-cation with Tyr264 and Tyr268. H-bonds of an amide group with Leu162 and Gln269 are observed (Figure 4A).

Molecular dynamics for the best docking pose of Indinavir in the active site predict over 35% of time interactions with Tyr268, Leu162, and Gln269 (Figure 4B). Additionally, the formation of a new interaction with a chiral OH of indinavir and Gln269 can be observed for 62% of the total simulation time. At the Y268 site, interactions between indinavir and the residues Asp164, Tyr268 and Gln269 are also observed. Additionally, Lys157 establishes a hydrogen bond interaction with an amide oxygen. The RMSD plot for the protein and the complex are shown in Figure 4C. The worst stabilization for Indinavir was at the H73 site. (See Appendix A). Based on these results, it can be concluded that Indinavir is a good anti-SARS-CoV-2 candidate to be evaluated in vivo as a PLpro inhibitor, with a potential affinity for the C111 and Y268 active sites.

### 2.3. Design and Proposal of New Compounds

Based on molecule 43 (the most active of the series) and the information suggested by the QSAR equation, we carried out a variety of structural modifications in order to create new molecules with promising activity. We take as a basis the structural pattern of the most active compounds of the QSAR series, which contain the aryl-thiophene fragment. We also consider the descriptors of the QSAR equation, from which we see that the presence of basic nitrogen is favorable for the activity. In Figure 5, the best designed compounds are presented, together with their pIC_50_ value predicted by the equation. All the compounds presented activities higher than the most active compound of the series studied (compound 43, pIC_50_ = 6.947). The compounds with the best predicted activity were Pred12 and Pred13, both with a pIC_50_ value = 9.194, exceeding compound 43 by more than 2 log units. In order to perform an in silico validation of these proposals, we carried out docking and molecular dynamics studies at the C111 active site, at the Y268 site and at the H73 site of PLpro. The docking scores for each compound are presented in Figure 5 (XP GScore). The docking score values ranged from −7.0 to −14.0 kcal/mol. The selection criteria for the best compounds were as follows. First, the compounds with activity predicted by the QSAR greater than 6.9 were selected. Next, the compounds with the best scores at each site were selected for the final molecular dynamic calculations.

According to the docking and dynamic studies, the presence of basic nitrogen atoms is highly favorable. This is consistent with the presence of negative residues, such as Asp164, Glu167, Asp302, Asp76, Glu70, Asp62, Asp12, and Glu67, in the studied cavities. The equation also suggests that aromatic rings are favorable, which is consistent with the presence of aromatic residues. According to the docking and dynamic studies, these aromatic residues (Try171, 264, 268 and 273) perform pi-stacking and pi-cation type interactions with the compounds. Because the presence of oxygen, fluorine, and chlorine atoms contributes negatively to biological activity, these atoms were avoided in the design of new compounds.

The Pred14 molecule stands out by having the best scoring in the Y268 and C111 binding sites (−12.364, −14.412 respectively). Pred14 also has the second better score in H73, close to the Pred10 value (−9.320 versus 9.547, respectively). Positively charged nitrogen atoms of Pred14 interact with Asp164 and Glu167 via hydrogen bond and ionic interactions, both at the active site (C111) and at the Y268 site (Figure 6A,B). In the docking of Pred14 with H73, only an ionic interaction with the Glu70 residue was observed. In molecular dynamics, RMSD analysis indicates that the complex stabilizes at both the active site (C111) and at Y268 allosteric site (Figure 6C), but stabilization was erratic at H73. As for the duration of the interactions, the lowest prevalence was with Leu162, with 35% of the duration of the interaction in the active site. In this site also, the prevalence of 96% of the terminal amine protonated with the amino acid Asp302 stands out, while in H73 no significant interactions are observed. The second compound, PRED15, presented good scoring in the active site (−13.666) and the Y268 site (−11.750), and showed stable dynamics over time, making it the second compound with the highest PLpro inhibitor potential.

## 3. Materials and Methods

### 3.1. Formulations of the QSAR Equation

A series of 113 molecules with biological activity against PLpro was collected from the literature [37,43,46,48,49,50,51,52,53,54,55,57]. The activity is reported as IC_50_ measured in the enzyme under the same conditions. The values were converted to pIC_50_ (−logIC_50_ on molar scale). A series of structural and physicochemical descriptors were calculated using the PADEL-Descriptors program.

A matrix of pIC_50_ values (dependent variable) and descriptors (independent variables) was constructed. A PLS statistical regression was then performed in Minitab, and 29 compounds with high residual value (residuals greater than 0.5 log units) were discarded. The final model was built with 113 molecules, of which 90 were used as training set (79.6%) and 23 as test set (20.4%). The distribution of the biological activity values for training, test and the complete set is presented in the Figure 7. The final model was built using the following 13 descriptors: ALogP (lipophilicity), ALogP^2^, AMR (molar refractivity), DPSA-3 (Di-polarity/polarizability, Size and Shape, and Atom-type counts descriptor), nAr (number of aromatic rings), nBasic (number of basic nitrogen atoms), nAcid (number of carboxylic acids), nF (number of fluorine atoms), nCl (number of chlorine atoms), nHBDon (number of hydrogen bond donor atoms), nO (number of oxygen atoms), TopoPSA (topological polar surface area) and VABC (van der Waals volume with Atomic and Bond Contributions). Finally, we proceeded to execute the statistical validations of the model by calculating r^2^_test_, r^2^_rand-test_, q^2^_F1_, q^2^_F2_, q^2^_F3_, CCC, MAE and Y-random test.

The resulting equation was used to predict the biological activity (pIC_50_) of reported antivirals and for the design of new molecules.

### 3.2. Molecular Docking

Molecular dockings were performed using the native PLpro protein obtained from the Protein Data Bank, PDB = 7NFV, with a resolution of 1.42 Å. The protein was prepared using the Protein preparation Wizard module implemented in Maestro. All ions and molecules that do not belong to the protein were removed, except the Zinc atom. The ionization states of each atom were determined at a pH of 7.0 ± 0.5 and the protein was minimized in an OPLS4 force field. On the other hand, the ligands were optimized using the LigPrep module. The most probable protonation states at pH 7.0 ± 0.5 for each molecule were determined using the Epik algorithm. Each compound was manually reviewed to guarantee the correct stereochemistry, as well as the correct state of protonation.

The induced fit docking was executed on the Glide module. Each ligand was docked in the 3 possible sites of PLpro. The grid centers for each case were defined around the following amino acids: (a) C111, for the catalytic site; (b) Y268, for the first allosteric site; and (c) H73 for the second allosteric site. The size of the box was 20 Å^3^ in each case.

### 3.3. Selection QSAR-Docking Criteria

The final analysis was carried out on two sets of molecules: antivirals reported in the literature, and molecules designed from our QSAR equation.

As the first filter criteria, those molecules with a pIC_50_ predicted by the QSAR greater than 6.9 were selected for the docking studies. The second filter consisted in that the first four molecules of each group with the best XP Glide Score for each of the three possible sites (C111, Y268, and H73), were subjected to molecular dynamics.

### 3.4. Molecular Dynamics

The best pose of each molecule was subjected to molecular dynamics using the Desmond module implemented in Maestro. The dynamics were executed using the NPT isothermal-isobaric ensemble. The calculation conditions were 310 K, 1.01325 bar, and 0.15 M NaCl. The total simulation time of each compound was 100 ns. 1000 Frames were recorded for each simulation.

## 4. Conclusions

In conclusion, a QSAR model was created for a series of 113 molecules with PLpro inhibitory activity against SARS-CoV-2. Good statistical values were presented by the equation (q^2^ = 0.770, r^2^ = 0.832, r^2^_test_ =0.721), and validation was performed externally by calculating various descriptors, including the Y-random test ($\bar{q_{Y}^{2}}$ = 0.0003). The main structure–activity relationships were derived from the equation, indicating that the presence of basic nitrogen atoms, aromatic rings, and hydrogen bonding groups is favorable for PLpro inhibitory activity, while the presence of oxygen, fluorine, and chlorine atoms is unfavorable. Several commercial antiviral drugs were evaluated with this information, revealing that indinavir is a good candidate. Through docking and molecular dynamics studies, it was confirmed that lasting interactions were established by the compound during simulation time with Asp164, Tyr264, Tyr268, Leu162, and Gln269, among the most relevant residues. Finally, the design and in silico evaluation of new synthetically feasible molecules were carried out. The compound Pred14, a derivative of aryl-thiophene without chiral centers, presented the best profile. Pred14 exhibited good activity predicted by QSAR (pIC_50_ = 7.966) and good scoring values in docking (−14.4 kcal/mol in the C111 active site). According to molecular dynamics simulations (100 ns), the main interactions of Pred14 and PLpro were with Asp164 and Glu167. Therefore, this model represents a contribution to the understanding of the structure–activity relationship for PLpro inhibitors, enabling the evaluation of the second use of commercial drugs and the design of new compounds with promising anti-SARS-CoV-2 activity. While it is true that our computational approach provides valuable insights, future research endeavors should prioritize experimental validation to further consolidate our findings and address the identified limitations.

## Figures and Tables

**Figure 1 pharmaceuticals-17-00606-f001:**
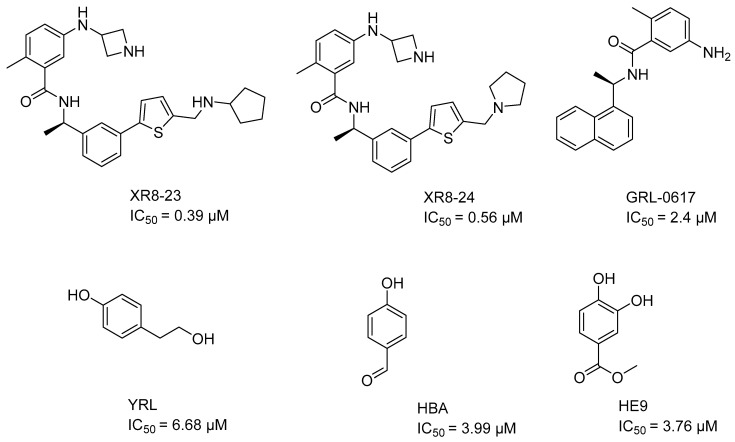
PLpro inhibitors reported in the literature. The activity values are in the micromolar and sub-micromolar range, therefore they are far from being able to be turned into commercial drugs. In addition, the presence of functional groups, such as aldehyde in HBA, makes them very toxic, or deficient in pharmacokinetics, due to their high solubility in water and rapid metabolization as in YRL and HE9.

**Figure 2 pharmaceuticals-17-00606-f002:**
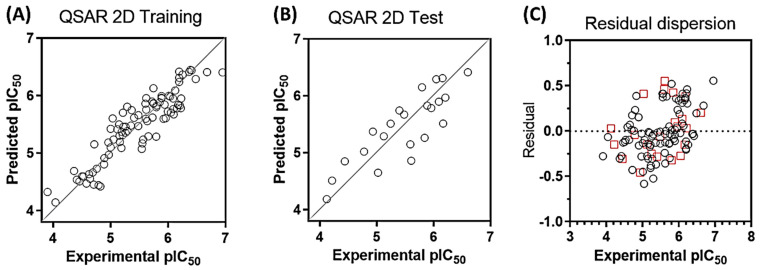
Plots of experimental versus predicted activity for the training set (**A**) and test set (**B**). Residual distribution plot (**C**). Black circles, training set, red squares, test set. All compounds are within a logarithmic unit of residual value (residual = pIC_50_ exp − pIC_50_ pred).

**Figure 3 pharmaceuticals-17-00606-f003:**
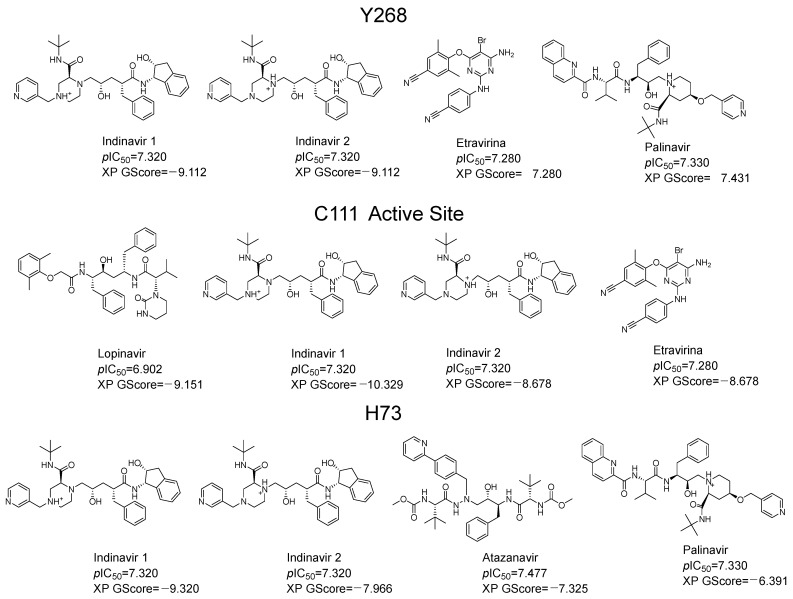
Predictive values of pIC_50_ of antivirals on the market calculated from QSAR equation. The docking scoring values (XP GScore) of each compound in its respective ionized form are shown for the tree binding sites (Y268, C111 and H73). In the case of Indinavir, the two possible protonation states at pH 7.0 are shown.

**Figure 4 pharmaceuticals-17-00606-f004:**
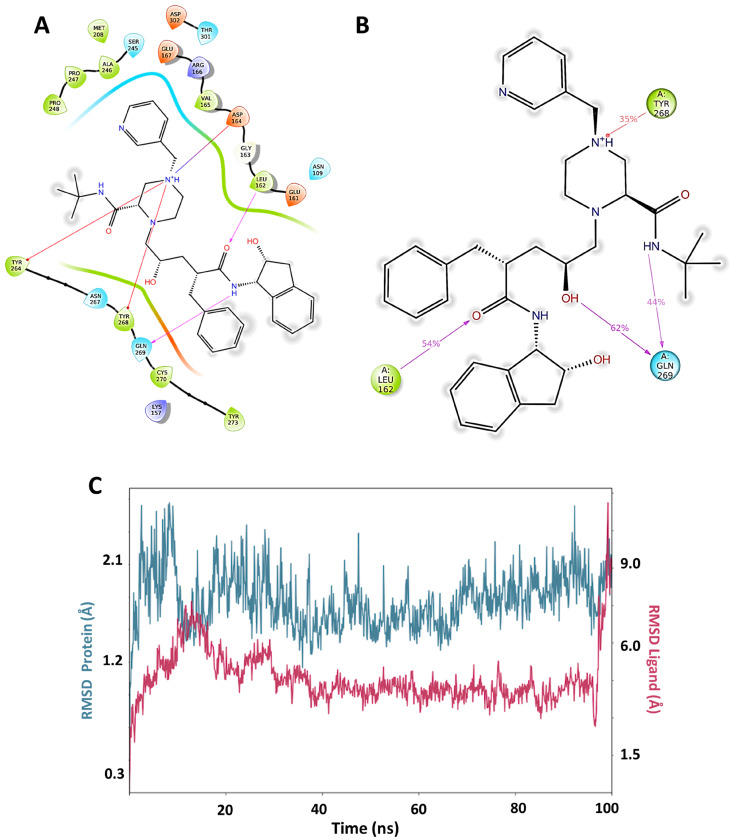
(**A**) Docking interactions for the best protonation state of Indinavir at the Y268 site. Residues are shown no more than 4 Å away. (**B**) Molecular dynamics result obtained from 100 ns of simulation for the pose shown in A. The dwell times of the interactions throughout the simulation are shown. A new relevant interaction can be seen between the OH group and Gln269. (**C**) RMSD graph for Indinavir and PLpro. From 20 ns onwards, a stabilization of the complex can be seen until the end of the simulation.

**Figure 5 pharmaceuticals-17-00606-f005:**
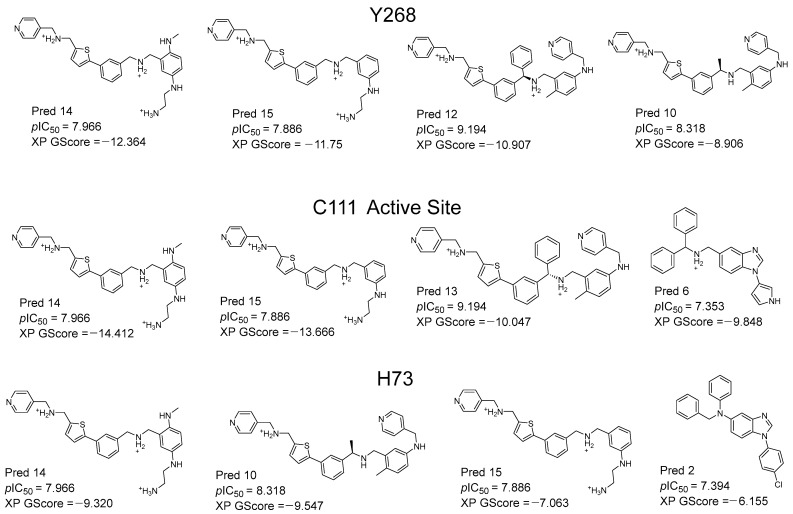
Structures of the best designed compounds and their predictive values of pIC_50_ according to the QSAR. The docking scoring values (XP GScore) of the best compounds in the three PLpro sites (Y268, C111 and H73) are also shown. Molecules are shown in the most favorable protonation state at pH 7.0.

**Figure 6 pharmaceuticals-17-00606-f006:**
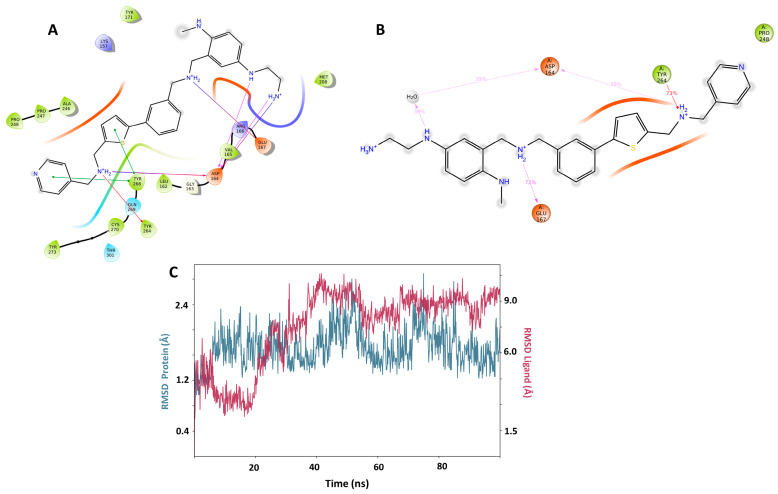
(**A**) Docking interactions for the best designed compound Pred14 at the Y268 site. Residues are shown no more than 4 Å away. (**B**) Molecular dynamics result obtained from 100 ns of simulation for the pose shown in A. The dwell times of the interactions throughout the simulation are shown. The docking interactions with Tyr264, Asp164, and Glu167 are preserved over time. (**C**) RMSD graph for Pred14 and PLpro. From 20 ns onwards, a stabilization of the complex can be seen until the end of the simulation.

**Figure 7 pharmaceuticals-17-00606-f007:**
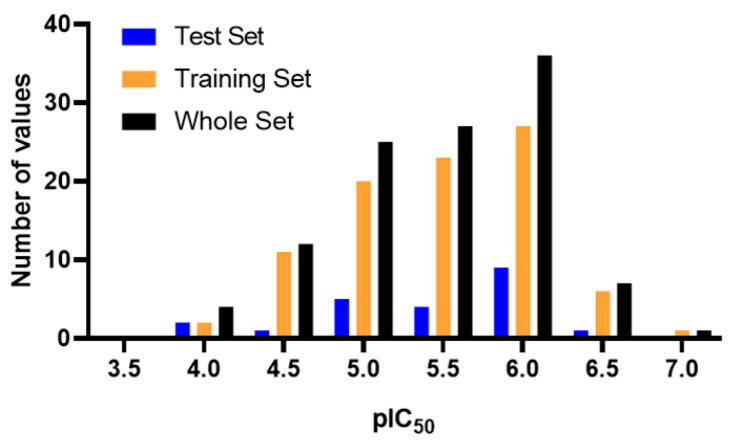
Histogram of frequency distribution of pIC_50_ values.

**Table 1 pharmaceuticals-17-00606-t001:** Statistical parameters for internal and external validation of the QSAR model.

Condition	Parameter	Threshold Value	Result
1	r^2^	high	0.8333
2	$q^{2}$	>0.5	0.770
3	$q_{F1}^{2}$	>0.7	0.817
4	$q_{F2}^{2}$	>0.7	0.817
5	$q_{F3}^{2}$	>0.7	0.984
6	MAE	Low	0.237
7	CCC	>0.85	0.905
8	$\bar{q_{Y}^{2}}$	<0.5	0.0003
9	$\bar{r_{Y}^{2}}$	<0.6	0.118
10	r^2^_test_	>0.6	0.721

q^2^ = the square of the LOO cross-validation (CV) coefficient; $q_{F1,F2,F3}^{2}$ are the coefficient for the variance explained in external prediction; MAE = Mean absolute error in fitting; CCC = Concordance correlation coefficient; $\bar{q_{Y}^{2}}$ and $\bar{r_{Y}^{2}}$ are the mean q^2^ and r^2^ for the Y-random test respectively; r^2^_test_ is the regression coefficient for the test set exclusively.

**Table 2 pharmaceuticals-17-00606-t002:** Experimental activities (pIC_50_ exp), predicted activities (pIC_50_ pred) and residual values (Res) for the set of molecules studied.

	pIC_50_				pIC_50_				pIC_50_		
Mol	Exp	Pred	Res	Mol	Exp	Pred	Res	Mol	Exp	Pred	Res
1	5.7932	5.2703	0.52	39	6.1938	6.3979	0.20	77	3.9031	4.1783	0.28
2	5.9586	5.7228	0.24	40	6.3872	6.4158	0.03	78 *	4.2157	4.3637	0.15
3	5.2596	5.5638	0.30	41	6.6778	6.3946	0.28	79 *	5.6073	5.0538	0.55
4	5.2218	5.6027	0.38	42	6.3665	6.3969	0.03	80	5.2924	5.3276	0.04
5	5.9957	5.6569	0.34	43	6.9469	6.3889	0.56	81	5.1938	5.3670	0.17
6	6.2218	5.7595	0.46	44 *	6.6021	6.3989	0.20	82	5.1549	5.2915	0.14
7	5.9208	5.5601	0.36	45 *	6.0915	5.9576	0.13	83 *	4.7773	4.8182	0.04
8	6.0969	5.6826	0.41	46	5.7447	5.8732	0.13	84	4.7077	4.5312	0.18
9	6.1549	5.8123	0.34	47 *	5.9586	5.9038	0.05	85	4.7905	4.5555	0.23
10	5.7959	5.8425	0.05	48	5.6383	5.6343	0.00	86	4.8928	5.0501	0.16
11	5.3665	5.7334	0.37	49	5.8539	5.9360	0.08	87	4.6253	4.7200	0.09
12	5.6198	5.7446	0.12	50	5.5436	5.1288	0.41	88	4.7595	4.7559	0.00
13	5.4089	5.4229	0.01	51	5.1612	5.1707	0.01	89	4.6882	4.6716	0.02
14	5.8861	5.8705	0.02	52	4.8239	4.4353	0.39	90	4.4123	4.6861	0.27
15 *	5.4815	5.5415	0.06	53 *	5.1249	5.2728	0.15	91	4.7160	5.1436	0.43
16	5.2147	5.6203	0.41	54	5.0000	5.4479	0.45	92	5.1487	5.4517	0.30
17	4.9706	4.9969	0.03	55	5.0000	5.1307	0.13	93 *	4.9314	5.3882	0.46
18	5.4815	5.4849	0.00	56	5.1192	5.3934	0.27	94	5.2899	5.8119	0.52
19	5.6198	5.9756	0.36	57 *	5.2441	5.4936	0.25	95	6.2076	5.7904	0.42
20	4.9626	5.2422	0.28	58	4.8861	4.7325	0.15	96	5.8013	5.7898	0.01
21 *	5.7959	6.1161	0.32	59	4.3645	4.6669	0.30	97 *	5.8861	5.7878	0.10
22	5.7212	5.8610	0.14	60	4.5258	4.6212	0.10	98	5.6073	5.8467	0.24
23	5.7447	6.0421	0.30	61	4.5058	4.6240	0.12	99	6.1805	5.8153	0.37
24	5.5528	5.1787	0.37	62	4.4584	4.5846	0.13	100	6.1739	5.8121	0.36
25	6.2291	5.9227	0.31	63 *	4.4389	4.7455	0.31	101	5.9393	5.8987	0.04
26	5.8861	5.9265	0.04	64 *	5.0223	4.6103	0.41	102 *	5.3990	5.6804	0.28
27	5.7447	5.7750	0.03	65	4.5800	4.5345	0.05	103	5.8861	5.9937	0.11
28 *	6.0458	6.3180	0.27	66	4.6421	4.5707	0.07	104	5.1878	5.2057	0.02
29	6.4089	6.4614	0.05	67 *	4.1209	4.0921	0.03	105	5.0511	5.1481	0.10
30	6.2518	6.4008	0.15	68	4.0400	4.1036	0.06	106	5.5654	5.1510	0.41
31	6.1249	6.0665	0.06	69	6.1938	6.3859	0.19	107	5.2596	5.3131	0.05
32	6.0132	5.8246	0.19	70 *	6.1675	5.7847	0.38	108	5.2132	5.3635	0.15
33	6.0915	5.7794	0.31	71 *	6.2076	6.1755	0.03	109	6.0506	5.6907	0.36
34	6.0362	6.0576	0.02	72	5.7305	5.6786	0.05	110	5.3546	5.4823	0.13
35	6.0862	6.0020	0.08	73	5.4841	5.6207	0.14	111 *	5.8416	5.4159	0.43
36	6.1938	6.3055	0.11	74	5.6126	5.6438	0.03	112	5.0353	5.6163	0.58
37 *	6.1549	6.3037	0.15	75 *	5.5918	5.1286	0.46	113 *	5.0888	5.1672	0.08
38	6.4815	6.2844	0.20	76	5.6576	5.2757	0.38				

(*) Test set compounds. pIC_50_ = −logIC_50_. The Res are absolute values.

## Data Availability

The dataset of the present study can be downloaded from Appendix A.

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
