# Peer review of "Arylamines QSAR-Based Design and Molecular Dynamics of New Phenylthiophene and Benzimidazole Derivatives with Affinity for the C111, Y268, and H73 Sites of SARS-CoV-2 PLpro Enzyme"

_pharmaceuticals, 2024, doi:10.3390/ph17050606_

Round 1

Reviewer 1 Report

Comments and Suggestions for Authors

In the manuscript entitled “QSAR-Based Design and Molecular Dynamics of New Phenylthiophene and Benzimidazole Derivatives with Affinity for the C111, Y268, and H73 Sites of SARS-CoV-2 PLpro Enzyme”, the author provided a QSAR model of a molecules with PLpro inhibitory activity of SARS-COV-2. Based on the formulated QSAR equation, they predicted the functional groups of the potential inhibitor molecule with positive and with negative effect to the inhibitory activity toward PLpro. Based on the QSAR model, they evaluated a commercial antiviral drug with the highest inhibitory activity, thereby wanting to identify a compound that has both antiviral and inhibitory activity. Finally, they carried out the design and in-silico evaluation of new synthetically feasible molecules with antiviral and inhibitory activities. In-silico evaluation is performed using appropriate molecular docking and molecular dynamics simulations.

The conceptualization of the investigations as well as the methodology approach is very correctly. The writing style is very clear and concise. The results are interesting, and presented correctly.

With everything mentioned in mind, I suggest publishing this review article after following changes, mostly of a technical nature:

1.       I suggest that the text in the Abstract and in the Conclusion be written in the passive voice, instead of the active voice. It is a common academic style of writing. Besides, by writing the text in the passive voice, it is avoided showing a personal relationship to the text and the results presented in the paper, and the focus is placed on the results themselves.

2.       In the text, there is a lot of informations with the following content: Error! Reference source not found. All of them should be removed and corrected.

3.       In the Abstract the number 2 nearby r, q and r should be written in the superscript (line 21).

4.       In the line 35, the presented year should be written correctly.

5.  In the sentence:  Another E3 domain is MDM2, which can monoubiquinate p53, which causes it to be exported to the cytosol, inhibiting apoptosis [23], which theoretically could favor cancer development, there are too much “which” words. This sentence should be written more stylishly and correctly.

6.       The name of test is Y-random test (The letter Y is written in a capital letter, and the word random is written without type errors), so it should be corrected (lines 132, 303 and 339)

7.       I suggest the authors to present the absolute residual values in Table 2, and not the true calculated residual values. Under my opinion, it is not particularly important whether the deviation of the predicted activities from the experimental ones is positive or negative.

8.       In the equation obtained from the 5-component PLS analysis some parts of the equation are written in red. Are there any special reason for that?

9.       There are some discrepancies between the descriptor labels in the equation and their explanation and spelling in the text that should be written uniformly:

·         in the equation: ALogp2 (p is in lowercase and 2 is in superscript), in the lines 154 and 295 it is written ALogP2 (The P is a capital letter and the 2 is in line with the other letters, normally.

·         in the equation: DPSA-3 is in Italic letter, and in the text it is Normal text

·         in the equation: nAr, in the text is nAR

·         in the equation there is nBasic value, but there are no explanation what it descriptor is.

·         In the text (lines 157 and 299) there are explanation of the nBr descriptor, but there are no nBr descriptor in the equation.

·         in the equation the number of hydrogen bond donor atoms is labelled nHBDON, but in the text, line 169 it is written nHBDon.

10.   In the line 200 there is: The_main , with an underscore between words.

Author Response

In the manuscript entitled “QSAR-Based Design and Molecular Dynamics of New Phenylthiophene and Benzimidazole Derivatives with Affinity for the C111, Y268, and H73 Sites of SARS-CoV-2 PLpro Enzyme”, the author provided a QSAR model of a molecules with PLpro inhibitory activity of SARS-COV-2. Based on the formulated QSAR equation, they predicted the functional groups of the potential inhibitor molecule with positive and with negative effect to the inhibitory activity toward PLpro. Based on the QSAR model, they evaluated a commercial antiviral drug with the highest inhibitory activity, thereby wanting to identify a compound that has both antiviral and inhibitory activity. Finally, they carried out the design and in-silico evaluation of new synthetically feasible molecules with antiviral and inhibitory activities. In-silico evaluation is performed using appropriate molecular docking and molecular dynamics simulations.

The conceptualization of the investigations as well as the methodology approach is very correctly. The writing style is very clear and concise. The results are interesting, and presented correctly.

Reply: Dear reviewer, thank you very much for your positive feedback. Then we provide a point-by-point response to all your comments. We have carried out all the corrections you provide us. We hope with these amendments our manuscript be now suitable for final publication. Best regards.

With everything mentioned in mind, I suggest publishing this review article after following changes, mostly of a technical nature:

I suggest that the text in the Abstract and in the Conclusion be written in the passive voice, instead of the active voice. It is a common academic style of writing. Besides, by writing the text in the passive voice, it is avoided showing a personal relationship to the text and the results presented in the paper, and the focus is placed on the results themselves.

Reply: We appreciate your helpful comment. The abstract and conclusion have been rectified and rewritten in the passive voice.

In the text, there is a lot of informations with the following content: Error! Reference source not found. All of them should be removed and corrected.

Reply: We sincerely apologize for the error. It occurred during the PDF conversion process on the platform. We have corrected it to ensure that the error does not occur again.

In the Abstract the number 2 nearby r, q and r should be written in the superscript (line 21).

Reply: The numbers were corrected and placed in superscript.

In the line 35, the presented year should be written correctly.

Reply: The year has been corrected.

In the sentence:  Another E3 domain is MDM2, which can monoubiquinate p53, which causes it to be exported to the cytosol, inhibiting apoptosis [23], which theoretically could favor cancer development, there are too much “which” words. This sentence should be written more stylishly and correctly.

Reply: The sentence has been enhanced and refined for clarity and coherence.

The name of test is Y-random test (The letter Y is written in a capital letter, and the word random is written without type errors), so it should be corrected (lines 132, 303 and 339).

Reply: Thank you for your comment. The name of the test has been corrected throughout the entire manuscript.

I suggest the authors to present the absolute residual values in Table 2, and not the true calculated residual values. Under my opinion, it is not particularly important whether the deviation of the predicted activities from the experimental ones is positive or negative.

Reply: According to reviewer’s comment we have updated Table 2 to display the Res absolute values.

In the equation obtained from the 5-component PLS analysis some parts of the equation are written in red. Are there any special reason for that?

Reply: Thank you for your comment. We have revised the equation to ensure that all elements are in black to prevent confusion.

There are some discrepancies between the descriptor labels in the equation and their explanation and spelling in the text that should be written uniformly: in the equation: ALogp2 (p is in lowercase and 2 is in superscript), in the lines 154 and 295 it is written ALogP2 (The P is a capital letter and the 2 is in line with the other letters, normally.

Reply: The exponent and the letter have been corrected.

in the equation: DPSA-3 is in Italic letter, and in the text it is Normal text

Reply: The term DPSA-3 has been corrected and putted in normal text.

in the equation: nAr, in the text is nAR

Reply: The nAr term has been corrected to nAr in all the manuscript.

in the equation there is nBasic value, but there are no explanation what it descriptor is.

Reply: The nBasic value has been described in the test (nBasic = number of basic nitrogen atoms)

In the text (lines 157 and 299) there are explanation of the nBr descriptor, but there are no nBr descriptor in the equation.

Reply: Thanks for the observation. The nBr term has been deleted from the main text (no nBr descriptor in the equation, indeed).

in the equation the number of hydrogen bond donor atoms is labelled nHBDON, but in the text, line 169 it is written nHBDon.

Reply: The nHBDon term has been uniformed in all the text.

  1. In the line 200 there is: The_main , with an underscore between words.

Reply: The “_” has been deleted.

Reviewer 2 Report

Comments and Suggestions for Authors

In this manuscript, the authors present a QSAR study of literature-based inhibitors of SARS-CoV-2 PLpro, followed by further evaluation through docking and molecular dynamic simulations. However, the manuscript requires careful polishing and revision before submission for peer review, with several areas for improvement.

1. Title Enhancement: The title should be refined to accurately reflect the content of the manuscript. Currently, it mentions a focus on "Phenylthiophene and Benzimidazole Derivatives" in the QSAR study, but lacks detailed information about the structural features common to the compounds used in QSAR analysis.

2. Updated References: The introduction section lacks references from recent years, with all citations predating 2023. Therefore, some of the descriptions in the Introduction section are not comprehensive. For example, in lines 41-42, it is obvious that there are at least 3 antiviral drugs/drug combinations on the market for the treatment of COVID-19. The authors should update the most recent references as this is a rapidly developing area.

3. Error Correction: Numerous errors and typos are present throughout the manuscript. Reference citing errors can be found in lines 82, 89, 118, 120-122, 125,127, 135, and others. In line 35, the phrase "At the end of 202" is missing the last number of years. Authors are responsible for checking the manuscript before submission.

4. Figure Resolution: The resolution of figures 4 and 6 is insufficient to see the details clearly. Higher-resolution versions of these figures should be provided to ensure clarity and readability for readers.

5. The docking and MD simulations are not good enough to verify the accuracy of the QSAR model. Authors should at least use PLpro inhibition assay to test some of the high-scoring compounds to get a solid conclusion.

Author Response

In this manuscript, the authors present a QSAR study of literature-based inhibitors of SARS-CoV-2 PLpro, followed by further evaluation through docking and molecular dynamic simulations. However, the manuscript requires careful polishing and revision before submission for peer review, with several areas for improvement.

Reply = Dear Reviewer, Thank you very much for your insightful comments. We sincerely appreciate the valuable feedback you have provided. A thorough point-by-point response has been provided for each observation. We have carefully revised our manuscript in accordance with your suggestions. Consequently, we believe that the paper is now well-prepared for publication. Best regards.

  1. Title Enhancement: The title should be refined to accurately reflect the content of the manuscript. Currently, it mentions a focus on "Phenylthiophene and Benzimidazole Derivatives" in the QSAR study, but lacks detailed information about the structural features common to the compounds used in QSAR analysis.

 Reply: We have updated the title adding information about the common feature of the data set.

  1. Updated References: The introduction section lacks references from recent years, with all citations predating 2023. Therefore, some of the descriptions in the Introduction section are not comprehensive. For example, in lines 41-42, it is obvious that there are at least 3 antiviral drugs/drug combinations on the market for the treatment of COVID-19. The authors should update the most recent references as this is a rapidly developing area.

 Reply: Dear reviewer we have updated our manuscript with more recent articles including from year 2024.

  1. Error Correction: Numerous errors and typos are present throughout the manuscript. Reference citing errors can be found in lines 82, 89, 118, 120-122, 125,127, 135, and others. In line 35, the phrase "At the end of 202" is missing the last number of years. Authors are responsible for checking the manuscript before submission.

 Reply: We sincerely apologize for the error. It occurred during the PDF conversion process on the platform. We have corrected it to ensure that the error does not occur again.

  1. Figure Resolution: The resolution of figures 4 and 6 is insufficient to see the details clearly. Higher-resolution versions of these figures should be provided to ensure clarity and readability for readers.

 Reply: Dear Reviewer, we sincerely apologize for the low-resolution images. Our original figures are high-resolution, but during the pdf conversion process on the platform, some images lack resolution. To ensure the high-res images are in the final version of our manuscript, we have uploaded the images as separate files to the platform.

  1. The docking and MD simulations are not good enough to verify the accuracy of the QSAR model. Authors should at least use PLpro inhibition assay to test some of the high-scoring compounds to get a solid conclusion.

Reply: We appreciate the reviewer's comments and suggestion regarding the validation of our QSAR model. We acknowledge the importance of validating computational results with additional experiments to strengthen our conclusions. However, we would like to emphasize that our study focused solely on computational methods due to resource and time constraints.

We understand that PLpro inhibition assays could provide additional data to support our predictions. However, it's worth noting that experimental assays involve significant costs and require appropriate infrastructure that was not available to us at the time of the research.

Nevertheless, we have acknowledged the limitations of our approach in our work and have been transparent about the need for further experimental validation to confirm our predictions. We appreciate the reviewer's suggestion and will seriously consider the possibility of conducting experimental assays in future research to support our computational findings (Conclusion section).

Round 2

Reviewer 2 Report

Comments and Suggestions for Authors

All the issues have been addressed appropriately.